# DREAM: Decoupled Reinforcement Learning with Reward Measurement for Large Language Model Test-time Training

## Abstract

This paper studies the problem of large language model (LLM) test-time training, which aims to enhance the reasoning ability of LLMs via unlabeled test data. Recent works usually utilize majority voting to infer the labels of samples to guide the reinforcement learning process, which could be inaccurate and biased with potential error accumulation. Towards this end, we propose a novel approach named Decoupled Reinforcement Learning with Reward Measurement (DREAM) for LLM test-time training. The core of our proposed DREAM is to decouple the reward estimation from reinforcement learning with enhanced calibration. In particular, our DREAM trains an LLM-based calibration model which takes both questions and answers as input, and outputs the calibration scores. To mitigate overconfident results, the judge model is trained by simulating on an independent reference dataset with positive and negative pairs. The reference-based calibration scores would be incorporated into voting-based reward estimation to reduce the potential biases, which enhance reliable test-time training. Extensive experiments on benchmark datasets validate the superiority of the proposed DREAM in comparison with competing baselines.

## 1 Introduction

Large language models (LLMs) (Zhao et al., 2024; Chang et al., 2024; Kasneci et al., 2023; Naveed et al., 2025) have become the backbone of modern natural language processing (NLP) (Khurana et al., 2023; Feuerriegel et al., 2025; Frenda et al., 2025), powering a wide range of downstream applications (Thirunavukarasu et al., 2023; Kaddour et al., 2023; Nie et al., 2024; Guo et al., 2024; Yan et al., 2025). Beyond their scale and architecture, the effectiveness of LLMs crucially depends on their training paradigm (Minaee et al., 2024; Wang et al., 2025; Dong et al., 2025), which can be broadly divided into *pre-training* (Ma et al., 2024) and *post-training* (Zosa et al., 2025; Kumar et al., 2025). While pre-training builds general-purpose capabilities from large-scale corpora, post-training emphasizes aligning LLMs with human preferences and adapting them to task-specific requirements, making it a critical stage for improving reasonability and reliability.

Among post-training paradigms for large language models (Lai et al., 2025; Lv et al., 2025; Tie et al., 2025), test-time reinforcement learning (Odena et al., 2017; Qu et al., 2025) has emerged as a promising strategy to enhance reasoning and generalization on complex tasks (Ge et al., 2023; Han et al., 2024) and unseen knowledge (Farquhar et al., 2023; Li et al., 2025). Unlike conventional reinforcement learning that requires labeled data, TTRL (Zuo et al., 2025) leverages unlabeled test inputs and majority voting based rewards to refine policies online without explicit supervision. Early approaches exploit consistency-based signals, such as self-consistency across multiple reasoning paths, to generate pseudo-rewards for stable adaptation (Wang et al., 2022; Ghosh et al., 2024), while subsequent work integrates calibration and uncertainty modeling to mitigate overconfident or biased feedback (Stangel et al., 2025; Park et al., 2025). Extensions of TTRL combine it with test-time training (TTT) (Sun et al., 2020; Osowiechi et al., 2024), introducing auxiliary self-supervised objectives to improve robustness under distribution shifts (Zhang et al., 2025d), and task-specific designs, such as leveraging region consistency in GUI grounding (Du et al., 2025), further demonstrate its adaptability. Collectively, these advances establish test-time reinforcement learning as a

flexible and rapidly evolving framework that unifies self-supervision, calibrated reward estimation, and reinforcement learning to enhance LLMs' reasoning, robustness, and task adaptability.

Despite the advantages of test-time reinforcement learning, several challenges remain when handling unlabeled data. For instance, TTRL (Zuo et al., 2025) derives rewards by aggregating multiple outputs via majority voting. However, such voting may yield inaccurate rewards that deviate from the ground-truth labels, thereby introducing negative feedback into the subsequent reinforcement learning process. These observations raise a critical question: ❶ *How can reinforcement learning be enhanced with reliable rewards in the absence of explicit supervision?* More importantly, reward generation and policy updates are inherently coupled within the reinforcement learning paradigm. Consequently, inaccurate rewards can lead to biased policy optimization, which in turn compounds generation errors and creates a vicious cycle during the post-training process. This concern motivates a second key question: ❷ *Can reward estimation be decoupled from policy optimization to mitigate bias accumulation in post-training?*

To address these challenges, we propose a simple yet effective framework DREAM for enhancing adaptation on unlabeled test data by decoupling reward estimation from reinforcement learning in LLM test-time training. Specifically, DREAM constructs an independent labeled reference dataset to train an auxiliary judge model, which is employed to estimate the reliability of model outputs. Built upon this, we design a hybrid reward calibration module that incorporates reliability scores predicted by the frozen judge model to produce final rewards. Subsequently, these calibrated rewards are leveraged to perform unbiased test-time reinforcement learning, thereby promoting generalizability when handling diverse inputs in unsupervised scenarios. To demonstrate the reliability and effectiveness of DREAM, we further conduct extensive experiments on benchmark datasets, showing outstanding advantages over state-of-the-art baseline approaches.

To better elucidate the contributions of DREAM, we summarize them as follows:

❶ **Novel Perspective.** To the best of our knowledge, we are the first to decouple reward estimation from reinforcement learning for unlabeled test data during the test-time training phase.

❷ **Unified Framework.** To enhance reinforcement learning without labeled data, we introduce an auxiliary module trained on independent reference data to estimate output reliability, which is subsequently leveraged to calibrate the final reward for policy optimization.

❸ **Empirical Validation.** We demonstrate the effectiveness of DREAM through extensive experiments on benchmark datasets against several baselines, showing its superiority empirically.

## 2 RELATED WORK

### 2.1 LLM POST-TRAINING

While *pre-training* provides a general foundation by learning from large-scale corpora, *post-training* aligns LLMs with specific knowledge, enhancing reasoning capabilities and refining outputs through target feedback (Kumar et al., 2025). Furthermore, classic paradigms for *post-training* include: (i) *Test-time scaling (TTS)*, which improves inference performance by allocating additional computational resources during reasoning (e.g. sequential scaling (Madaan et al., 2023; Brinkmann & Bizer, 2025) and parallel sequential (Korikov et al., 2025; Zhou et al., 2025)); (ii) *Fine-tuning*, which adapts the pretrained model to specific downstream domains to enhance task alignment Zhang et al. (2024); and (iii) *Reinforcement learning (RL)*, which optimizes models using reward signals derived from human preferences or other evaluative feedback (Laleh & Ahmadabadi, 2024; Yu et al., 2025; Hu et al., 2025). *Test-time scaling* improves inference performance without updating model parameters but incurs higher computational cost and may generalize poorly (Zhang et al., 2025c). *Fine-tuning* (Pletenev et al., 2025) enables strong task specialization but risks overfitting and requires high-quality labeled data. In contrast, *reinforcement learning* provides a flexible framework for post-training optimization, leveraging reward signals to enhance task alignment even under limited supervision. This flexibility motivates our focus on RL in the work.

### 2.2 REINFORCEMENT LEARNING IN LLMS

Reinforcement learning (RL) (Chen et al., 2025b; Dang & Ngo, 2025) has become a mainstream post-training paradigm for aligning large language models (LLMs) with human preferences (Zhang et al., 2025a; Yu et al., 2025; Hariharan, 2025), thereby enhancing their reasoning ability and adap-

tive generalization during inference. For instance, Proximal Policy Optimization (PPO) (Schulman et al., 2017) performs policy updates through multiple epochs of stochastic gradient ascent and serves as a cornerstone of Reinforcement Learning with Human Feedback (RLHF) (Bai et al., 2022; Chaudhari et al., 2024). Beyond on-policy methods (Mroueh et al., 2025), several off-policy approaches (e.g., Implicit Language Q-Learning (ILQL) (Snell et al., 2023) and VerifierQ (Qi et al., 2024)) exploit offline datasets or auxiliary verifier signals to improve efficiency and reasoning reliability. More importantly, DeepSeek-R1 (Guo et al., 2025) introduces Group Relative Policy Optimization (GRPO) (Zhang et al., 2025b; Sane, 2025), which extends LLMs' reasoning capabilities through outcome-based relative rewards. Despite these advances, most RL-based post-training (Sun & van der Schaar, 2025) frameworks rely on reliable supervised signals from humans or AI models, limiting their versatility when such labels are unavailable (Wang et al., 2024; Chen et al., 2025a;b). To address this challenge, Test-time Reinforcement Learning (TTRL) (Zuo et al., 2025) employs majority voting to derive consistency-based rewards, enabling reinforcement learning directly on unlabeled test data. However, coupling output generation and reward evaluation often leads to over-confident predictions and biased reinforcement learning. This motivates the need to decouple reward estimation from policy optimization for more robust test-time training.

## 3 THE PROPOSED DREAM

**Problem Definition.** In this work, we address the problem of test-time training for large language models (LLMs) using unlabeled test data. Formally, let $\mathcal{D}_U = \{x_i\}_{i=1}^{N_U}$ denote the set of test-time inputs without ground-truth labels, where $x_i \in \mathcal{X}$ belongs to the input space. Let $\pi_\theta(\cdot \mid x)$ represents the base policy model, which could generate candidate outputs $\{y_i\}$ according to $y_i \sim \pi_\theta(y \mid x)$ for a given input $x$. Since the true labels are unavailable, our objective is to design a reliable reward estimation module $R(\cdot)$ and to leverage the resulting calibration scores to optimize the policy model $\pi_\theta$ within a reinforcement learning framework.

### 3.1 DREAM: DECOUPLING REWARD ESTIMATION AND REINFORCEMENT LEARNING

To provide reliable rewards for test-time training with unlabeled data, our DREAM framework decouples reward estimation from reinforcement learning to generate enhanced calibration scores. Specifically, DREAM first generates multiple outputs from the policy model via repeated sampling, which are then employed to construct rewards for the unlabeled data through majority voting. To mitigate overly confident reward signals, we additionally train a judge model using a simulated independent reference dataset with supervised labels, which identifies whether the generated outputs are reliable. The calibration scores produced by the judge model are subsequently leveraged to further refine the rewards, thereby enabling more reliable test-time training. A detailed overview of the DREAM pipeline is presented in Figure 1.

### 3.2 CALIBRATION MODEL OPTIMIZATION WITH REFERENCE DATA

For unlabeled data, directly incorporating rewards into reinforcement learning (Zuo et al., 2025) may lead to biased and overconfident outcome during test-time training. To address this challenge, we introduce an additional model to calibrate the rewards, thereby providing more reliable evaluations based on an established reference data. More specifically, we construct a reference dataset $\mathcal{D}_L = \{(x_i, y_i)\}_{i=1}^{N_L}$, where $x_i$ denotes a question and $y_i$ represents its corresponding answer. This dataset is then leveraged to independently train a judge model. For each input $x_i$, we sample $K$ candidate outputs from the policy model $\pi_\theta(\cdot \mid x_i)$, forming a candidate set denotes as:

$$\mathcal{O}_i = \{\hat{y}_{i,k} \mid \hat{y}_{i,k} \overset{\text{i.i.d.}}{\sim} \pi_\theta(\cdot \mid x_i), 1 \le k \le K\}, \tag{1}$$

where $\hat{y}_{i,k}$ are independently and identically distributed (i.i.d.) samples draw from the policy model $\pi(\cdot \mid x_i)$. These outputs are then passed through a canonical answer extraction function $C$, which generates binary labels indicating whether each sampled output matches the ground-truth answer:

$$\mathcal{S}_i = \{s_{i,k}\}_{k=1}^K = \{\mathbb{I}(C(\hat{y}_{i,k}) = y_i) \mid \hat{y}_{i,k} \in \mathcal{O}_i\}, \tag{2}$$

where $s_{i,k}$ is the binary label representing the consistency with the ground-truth label, and $\mathbb{I}$ is the indicator function. Built upon these formulation, the judge model $F_\phi$ is trained to learn a mapping

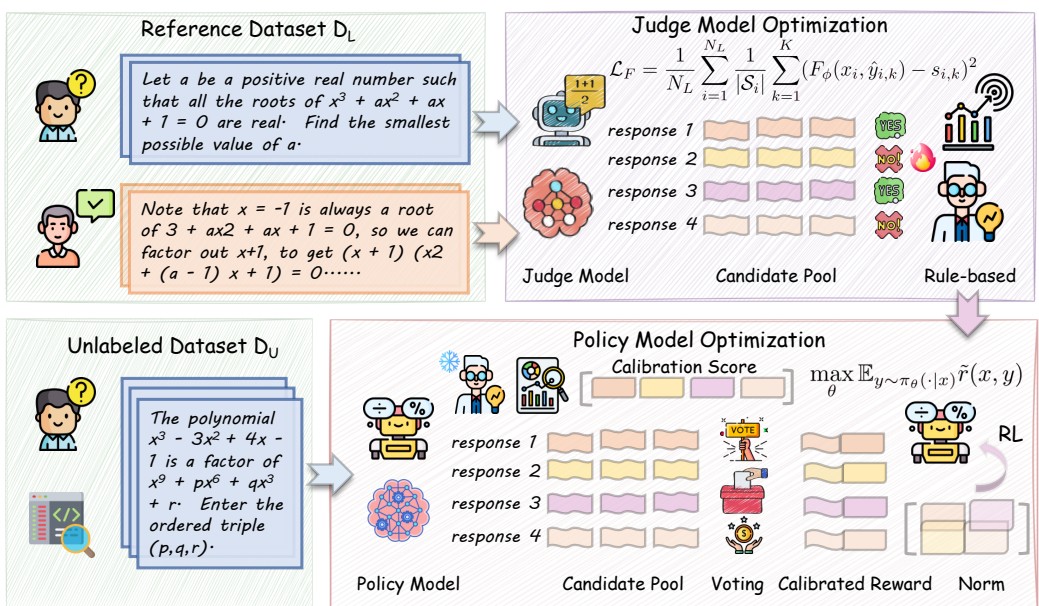

Figure 1: Overview of DREAM. The framework is composed of two optimization modules: judge model optimization and policy model optimization.

from $(x_i, \mathcal{O}_i)$ to $\mathcal{S}_i$, i.e., $F_\phi : (x_i, \hat{y}_{i,k}) \to s_{i,k}$. The parameters of $F_\phi$ are then optimized with a mean squared error (MSE) regression loss:

$$\mathcal{L}_F = \frac{1}{N_L} \sum_{i=1}^{N_L} \frac{1}{|\mathcal{S}_i|} \sum_{k=1}^{K} (F_\phi(x_i, \hat{y}_{i,k}) - s_{i,k})^2, \tag{3}$$

where $F_\phi(x_i, \hat{y}_{i,k})$ denotes the calibration score predicted by the judge model, and $s_{i,k}$ is the binary alignment score. By incorporating these additional signals, the judge model gains the ability to assess the reliability of generated outputs, thereby serving as an auxiliary module to refine reward estimation for unlabeled data during test-time training.

## 3.3 REWARD ENHANCEMENT THROUGH CALIBRATION

Generally, the policy model $\pi_\theta$ is optimized using a supervised reward function to guide reinforcement learning. However, in the absence of supervisory signals for unlabeled data, direct reward-based optimization becomes unreliable and overconfident. To tackle this, we propose a hybrid reward enhancement module that incorporates calibration via a previously trained judge model, enabling more effective utilization of rewards under unsupervised conditions.

**Voting-Based Consistency Reward.** To generate rewards that enhance reinforcement learning, we first construct a candidate set by repeatedly sampling from the policy model $\pi_\theta$. For the unlabeled dataset $\mathcal{D}_U = \{x_i\}_{i=1}^{N_U}$, an answer set $\mathcal{O}_i = \{\hat{y}_{i,k}\}_{k=1}^{K}$ is obtained for each input $x_i$ following Eq. 1. Subsequently, we apply majority voting (Zhou et al., 2024; Zhao et al., 2025) to evaluate the consistency within the answer set, formulated as:

$$r_{i,k}^v = \mathbb{I}(\hat{y}_{i,k} = \tilde{y}_i), \quad \text{where} \quad \tilde{y}_i = \text{MV}(\mathcal{O}_i) \quad \text{and} \quad 1 \le k \le K. \tag{4}$$

Here $K$ denotes the number of sampled outputs, and $\text{MV}(\mathcal{O}_i)$ represents the consensus output determined by the majority voting aggregation strategy. The indicator $r_{i,k}^v$ thus defines a consistency reward for each output in the candidate set $\mathcal{O}_i$. This approach allows us to derive intrinsic rewards in the absence of supervised signals, providing guidance for the optimization of the policy model.

**Calibration for Reward Reliability.** Recalling that we have trained a judge model $F_\phi$ to assess the confidence of each output individually. By integrating this confidence assessment component, we can decouple pseudo-label generation from quality evaluation on the unlabeled dataset. This

paradigm provides more reliable rewards to enhance the policy model through reinforcement learning during test-time training. Accordingly, the final reward for each output is obtained by combining the voting-based reward with the calibration score from the judge model:

$$r_{i,k} = r_{i,k}^v + s_{i,k}, \quad \text{where} \quad 1 \le i \le N_U, \quad 1 \le k \le K. \tag{5}$$

This approach decouples pseudo-labels from confidence estimation, promoting correct answers while penalizing incorrect ones, thereby mitigating overconfidence in erroneous outputs.

### 3.4 UNBIASED TEST-TIME REINFORCEMENT LEARNING

To enable unbiased test-time reinforcement learning, DREAM incorporates rewards that combine voting-based consistency with calibration scores for enhanced reliability.

**Group-wise Reward Normalization.** Group Relative Policy Optimization (GRPO) (Shao et al., 2024) optimizes policies by comparing the relative strengths of different responses within a group, which reduces both memory usage and computational overhead. Motivated by this, we normalize rewards using a candidate set $\mathcal{O}_i$. Specifically, given an output $\hat{y}_{i,k} \in \mathcal{O}_i$ with its associated reward $r_{i,k}$, the normalized score is computed as:

$$\tilde{r}_{i,k} = \frac{r_{i,k} - \mu_i}{\sigma_i}, \quad \text{where} \quad \mu_i = \frac{1}{K} \sum_{k=1}^{K} r_{i,k} \quad \text{and} \quad \sigma_i = \sqrt{\frac{1}{K} \sum_{k=1}^{K} (r_{i,k} - \mu_i)^2}. \tag{6}$$

This reward normalization provides a more stable and comparable signal, which can effectively guide the optimization of the policy model.

**Policy Model Optimization.** Here, we leverage the previously generated rewards to optimize the policy model using unlabeled data during the test-time training process. The reinforcement learning objective is to maximize the expected reward, formulated as:

$$\max_\theta \mathbb{E}_{y \sim \pi_\theta(\cdot|x)} \tilde{r}(x, y), \tag{7}$$

where $\tilde{r}(x, y)$ represents the calibrated reward according to Eq. 6. What's more, the parameters $\theta$ are then updated via gradient ascent:

$$\theta \leftarrow \theta + \gamma \cdot \mathbb{E}_{y \sim \pi_\theta(\cdot|x)} \tilde{r}(x, y), \tag{8}$$

in which $\gamma$ represents the learning rate. This update strategy provides a robust reward estimation by incorporating auxiliary calibration from an independent judge model, thereby enhancing the effective utilization of unlabeled data during test-time training. In conclusion, DREAM demonstrates significantly improved generalizability when handling diverse inputs under unsupervised scenarios.

### 3.5 TRAINING PROCEDURE OF DREAM

To enable better test-time training with unlabeled data, DREAM carefully decouples reward estimation from reinforcement learning, allowing the model to leverage unlabeled inputs without introducing overly confident or biased updates. The process proceeds in two interlinked stages, each designed to ensure reliable reward signals and stable policy optimization. (i) ***Judge model optimization.*** In the first stage, DREAM constructs an auxiliary judge model that serves as a reliable reward estimator. To provide supervision for the judge model, we generate a set of additional math questions with known ground-truth answers. For each question, the policy model is queried multiple times (e.g., $K = 4$) to produce diverse candidate solutions, capturing the inherent stochasticity and variability of the model outputs. Each candidate output is then evaluated against the corresponding ground-truth answer using rule-based verification. Outputs that correctly solve the question are grouped into positive pairs, while incorrect outputs form negative pairs. The labeled pairs are subsequently used to train the judge model using a supervised learning objective, allowing it to effectively assign reliability scores to candidate outputs. (ii) ***Policy model optimization.*** For a given math question, we first sample multiple outputs from the policy model to construct a candidate set $\mathcal{O}_i$, which captures the range of plausible solutions that the policy may generate. Each candidate in $\mathcal{O}_i$ is then evaluated by the judge model, which produces a calibration score reflecting its estimated quality. These scores are combined with a voting-based reward scheme, which are then refined by the calibration scores. Finally, these calibrated rewards are used to perform reinforcement learning updates

---

**Algorithm 1:** Test-time Training Procedure of DREAM

---

**Input:** Reference dataset $\mathcal{D}_L = \{(x_i, y_i)\}_{i=1}^{N_L}$ and policy model $\pi_\theta$
**1 for** iteration $i \leftarrow 1$ **to** $N_L$ **do**                              // Judge model optimization
**2**     Generate a candidate set $\mathcal{O}_i$ for each input $x_i$ using $\pi_\theta$ according to Eq. 1
**3**     Compute the matching score between $\hat{y}_{i,k} \in \mathcal{O}_i$ and the ground-truth $y_i$ by Eq. 2
**4**     Optimize the judge model $F_\phi$ using the MSE loss defined in Eq. 3
**5 end**
**Output:** Trained judge model $F_\phi$                    // Calibration for reward reliability
**Input:** Unlabeled test dataset $\mathcal{D}_U = \{x_i\}_{i=1}^{N_U}$ and the frozen judge model $F_\phi$
**6 for** iteration $i \leftarrow 1$ **to** $N_U$ **do**                              // Policy model optimization
**7**     Generate a candidate set $\mathcal{O}_i$ for each input $x_i$ using $\pi_\theta$ according to Eq. 1
**8**     Estimate the calibration score $s_{i,k}$ for each candidate $\hat{y}_{i,k}$   // Predict with judge model
**9**     Compute the voting-based reward $r_{i,k}^v$ within $\mathcal{O}_i$ using Eq. 4
**10**    Derive the calibrated reward $r_{i,k}$ according to Eq. 5
**11**    Normalize the group-wise reward $\tilde{r}_{i,k}$ using Eq. 6
**12**    Optimize $\pi_\theta$ by Eq. 7 and Eq. 8.
**13 end**
**Output:** Trained policy model $\pi_\theta$         // Optimize the policy with calibrated rewards

---

on the policy model, allowing it to improve its performance iteratively while accounting for the judge's assessment. The entire process, including both the judge and policy model optimizations, is formally summarized in Algorithm 1.

## 4 EXPERIMENT

### 4.1 EXPERIMENTAL SETUP

**Backbone Models.** To verify the generalizability of our reward score calibration, we conduct experiments on a broad range of backbone models, encompassing both base and instruct models at different scales. Specifically, we select Qwen2.5-Math-1.5B (Yang et al., 2024), Qwen2.5-Math-7B (Yang et al., 2024), Qwen2.5-7B (Qwen et al., 2025), LLaMA3.1-8B-Instruct (Dubey et al., 2024), and DeepSeek-R1-7B-Instruct (Guo et al., 2025) as backbone models.

**Evaluation Setup.** We evaluate DREAM on three challenging mathematical reasoning benchmarks: AIME 2024, AMC, and MATH-500. For better evaluation, we adopt avg@k, maj@k, and pass@k to evaluate the experimental results. Following the practice of (Zuo et al., 2025), we generate responses with non-zero temperature sampling with a temperature of 0.6 and a top-$p$ value of 0.95 unless specified. The maximum generation length is set to 3,072 tokens. We prompt the model to perform step-by-step reasoning and to present the final answer enclosed within "\boxed{}".

**Baselines.** Our primary baseline for comparison is TTRL (Zuo et al., 2025), the state-of-the-art method for test-time reinforcement learning on unlabeled data. This direct comparison allows us to isolate and measure the performance gains attributable to our decoupled reward calibration mechanism. Additionally, we report the performance of the original backbone models without any test-time training to establish a performance floor and quantify the absolute improvement.

**Implementation Details.** We implement DREAM by applying GRPO (Shao et al., 2024) to the policy model, guided by our calibrated reward signals. For the policy model's hyperparameters, we use a cosine learning rate schedule with a peak value of $5 \times 10^{-7}$ and employ the AdamW optimizer. During the rollout phase, we sample 64 responses per prompt, which are then used for majority voting and scoring by the judge model. Subsequently, we downsample to 32 responses per prompt for the training batch. The maximum generation length is set to 3,072 tokens. The number of training episodes is adapted to the dataset size: 80 for AIME 2024, 30 for AMC, and 10 for MATH-500. All experiments were conducted on a cluster of 8 NVIDIA H20 GPUs.

**Reward Calibration Model.** A core component of our methodology is the judge model for reward calibration. As the proprietary datasets used for the pre-training and supervised fine-tuning (SFT) of

Table 1: Performance comparison with TTRL on various backbone models. We report avg@16 on AIME 2024, AMC, and MATH-500 datasets.

| Backbone | Model | AIME 2024 | AMC | MATH-500 | Avg. |
|---|---|---|---|---|---|
| *Math Base Models* | Qwen2.5-Math-1.5B | 7.5 | 29.5 | 33.8 | 23.6 |
| | w/ TTRL | 12.1$_{\uparrow 4.6}$ | 45.9 $_{\uparrow 16.4}$ | 68.2$_{\uparrow 34.4}$ | 42.1$_{\uparrow 18.5}$ |
| | w/ DREAM | **16.7**$_{\uparrow 9.2}$ | **46.8**$_{\uparrow 17.3}$ | **72.9**$_{\uparrow 39.1}$ | **45.5**$_{\uparrow 21.9}$ |
| | Qwen2.5-Math-7B | 13.5 | 36.5 | 46.5 | 32.2 |
| | w/ TTRL | **40.8**$_{\uparrow 27.3}$ | 67.0$_{\uparrow 30.5}$ | 83.4$_{\uparrow 36.9}$ | 63.7$_{\uparrow 31.5}$ |
| | w/ DREAM | 40.0$_{\uparrow 26.5}$ | **67.4**$_{\uparrow 30.9}$ | **84.5**$_{\uparrow 38.0}$ | **64.0**$_{\uparrow 31.8}$ |
| *Vanilla Base Model* | Qwen2.5-7B | 7.3 | 32.0 | 60.9 | 33.4 |
| | w/ TTRL | 23.3$_{\uparrow 16.0}$ | 51.8$_{\uparrow 19.8}$ | **81.1**$_{\uparrow 20.2}$ | 52.1$_{\uparrow 18.7}$ |
| | w/ DREAM | **23.3**$_{\uparrow 16.0}$ | **55.4**$_{\uparrow 23.4}$ | 80.1$_{\uparrow 19.2}$ | **52.9**$_{\uparrow 19.5}$ |
| *Instruct Model* | LLaMA3.1-8B-Instruct | 4.8 | 21.5 | 48.4 | 24.9 |
| | w/ TTRL | 10.0$_{\uparrow 5.2}$ | 32.3$_{\uparrow 10.8}$ | **61.4**$_{\uparrow 13.0}$ | 34.6$_{\uparrow 9.7}$ |
| | w/ DREAM | **13.3**$_{\uparrow 8.5}$ | **34.8**$_{\uparrow 13.3}$ | 59.6$_{\uparrow 11.2}$ | **35.9**$_{\uparrow 11.0}$ |
| | DeepSeek-R1-7B-Instruct | 1.7 | 16.0 | 43.1 | 20.3 |
| | w/ TTRL | 1.0$_{\downarrow 0.7}$ | 19.3$_{\uparrow 3.3}$ | 50.8$_{\uparrow 7.7}$ | 23.7$_{\uparrow 3.4}$ |
| | w/ DREAM | **1.7**$_{\uparrow 0.0}$ | **20.8**$_{\uparrow 4.8}$ | **51.5**$_{\uparrow 8.4}$ | **24.7**$_{\uparrow 4.4}$ |

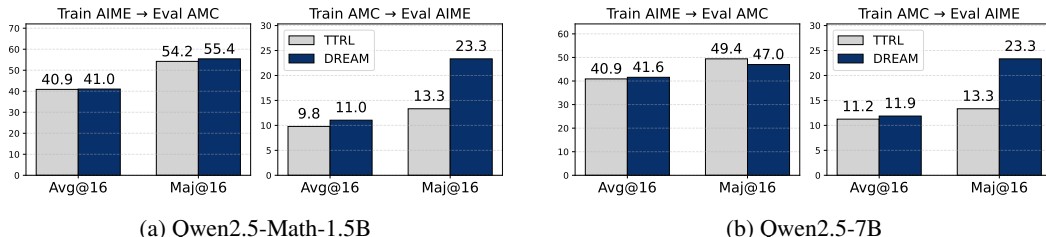

(a) Qwen2.5-Math-1.5B      (b) Qwen2.5-7B

Figure 2: Performance comparison of our DREAM and TTRL on out-of-distribution testing data. We select Qwen2.5-Math-1.5B as the backbone model and report avg@16.

the backbone models are often not accessible, we simulate a representative labeled dataset to train our reward calibration model. We utilize the training split of the MATH dataset for this purpose. For each question in the MATH training set (Hendrycks et al., 2024), we generate four candidate responses from the target policy model using a sampling temperature of 0.6 and a top-p of 0.9. These responses are then assigned a binary score based on a rule-based comparison against the ground-truth answer. This process yields a dataset of $<$ question, response, score $>$ tuples for training the judge model as a regression task. Further details on the judge model's training data distribution, architecture, and optimization are provided in Appendix A.

## 4.2 EXPERIMENT RESULTS AND ANALYSIS

**Reward calibration generally leads to better test-time RL training.** As shown in Table 1, DREAM mitigates the accumulation of reward noise caused by overconfidence during training by calibrating the voting-based reward with a decoupled reliability estimation, thereby demonstrating a positive average performance gain over the majority voting baseline on most of the scenarios. The benefits of our calibration mechanism are most pronounced for weaker models, where the initial reward signal from majority voting is inherently noisier. For example, DREAM achieves a significant improvement of 4.6 on AIME 2024 for the smaller Qwen2.5-Math-1.5B, as its judge model effectively filters erroneous positive rewards and mitigates overconfident optimization on flawed outputs. However, the impact of calibration can be nuanced. We observed a marginal performance decrease with DREAM on Qwen2.5-Math-7B for the AIME dataset. This could be attributed to our implementation choice for computational efficiency, where a single reward calibration model based on Qwen2.5-Math-1.5B was used across all experiments. This approach may be suboptimal when applied to larger, more capable models.

**Reward calibration generalizes well beyond the target task.** We evaluate the models trained with DREAM on out-of-distribution (OOD) test data to assess their generalization capabilities. Specif-

ically, we conduct cross-dataset validation between AIME 2024 and AMC. For each experiment, we perform test-time training on one dataset and evaluate the resulting model on the other one. As shown in Figure 2, the results on Qwen2.5-Math-1.5B and Qwen2.5-7B, measured by the avg@16 and maj@16, reveal that DREAM consistently improves upon both the backbone model and the TTRL baseline. This not only validates that the improvements from test-time RL are generalizable, but also indicates that the advantages of our method do not stem from overfitting to either the in-distribution test data or the reward calibration model.

**Analysis of the reward calibration model.** The effectiveness of DREAM, which calibrates reward scores by decoupling the reward reliability measurement process, is based on the hypothesis that for mathematical tasks, it is easier for an LLM to act as a judge model and determine the correctness of a solution than to generate the correct answer itself (Tan et al., 2025).

Table 2: Evaluation results of the reliability scores generated by the calibration model.

| Dataset | Accuracy | AUC | F1-score |
|---------|----------|------|----------|
| AIME 2024 | 84.58 | 0.820 | 49.32 |
| AMC | 73.57 | 0.821 | 72.21 |
| MATH-500 | 96.66 | 0.965 | 84.85 |

Therefore, we further evaluate the ability of our reward calibration model to measure response reliability for a given question. Following the pipeline used to construct the training set in Section 4.1, we built OOD test sets from the AIME 2024, AMC, and MATH-500 datasets using the same methodology. We binarize the continuous reliability scores from the reward calibration model at a 0.5 threshold into labels indicating whether a response is correct, and then evaluate it on this binary classification task using common metrics such as accuracy, AUC, and F1-score. As observed in Table 2, while the base Qwen2.5-Math-1.5B model achieves an avg@16 of only 7.5 on AIME 2024, our reward calibration model, after fine-tuning on a small dataset, reaches an accuracy of 84.58% and an AUC of 0.820.

Table 3: Ablation Study of DREAM's reward calibration process. In this table, "**MV**" represents majority voting, while "**RS**" denotes reliability score.

| Method | MV | RS | AIME | | AMC | |
|--------|-----|-----|-------|--------|--------|--------|
| | | | Avg@16 | Maj@16 | Avg@16 | Maj@16 |
| Qwen2.5-Math-1.5B | | | 7.5 | 20.0 | 29.5 | 42.2 |
| w/ TTRL | ✔ | | 12.1 | 20.0 | 45.9 | **53.0** |
| w/ Reliability Score | | ✔ | 11.9 | 20.0 | 44.0 | 51.8 |
| w/ DREAM | ✔ | ✔ | **16.7** | **23.3** | **46.8** | 51.8 |

**Ablation study of DREAM.** As shown in Eq. 5, our method, DREAM, uses the estimated reliability score $s_{i,k} \in (0,1)$ to calibrate the voting-based reward $r_{i,k}^v \in \{0,1\}$, rather than directly using $s_{i,k}$ as a fine-grained reward score. Therefore, we further explore the usage of the calibration score as the reward, with the results presented in Table 3. As observed, on Qwen2.5-Math-1.5B, using the calibration score alone for test-time RL training also yields a considerable improvement over the backbone model. However, it still slightly underperforms the majority voting reward of TTRL. In contrast, combining the two leads to a significant improvement. We attribute this to the fact that the performance of using $s_{i,k}$ alone as a reward is constrained by the capability of the reward calibration model itself. Incorrect estimations of the reliability score can introduce noise during training. More importantly, the reward calibration model is frozen during training and thus lacks the positive feedback property of majority voting, where the accuracy of reward estimation and model performance are mutually reinforcing. However, majority voting is prone to overconfidence, leading to the accumulation and amplification of errors from the early stages of training. Therefore, by using $s_{i,k}$ to correct the voting-based reward $r_{i,k}^v$ during training, DREAM combines the advantages of both approaches, achieving superior test-time RL performance.

**Comparison between internal and external reward calibration model.** To maintain a self-evolving framework, the reward calibration model should be *internal* for test-time RL training. Specifically, its construction process involves only the labeled data from the pre-training/SFT stages and the model itself, without relying on any external knowledge. In this setting, our method can be naturally adapted to any unsupervised RL scenario. However, if the focus is solely on test-time RL, we can introduce an *external* reward calibration model for reliability estimation. We conducted

a preliminary exploration of this approach, where we replaced the internal model with Qwen2.5-Math-PRM-7B.

Here, instead of using process-supervised rewards, we compute the reward directly at the entire response level. We performed tests similar to those in Table 2 and found that the quality of the reliability scores obtained this way is viable. The comparison between the internal and external calibration models is presented in Table 4. As shown, their performance is largely on par across three models and two

Table 4: Comparison between internal and external reward calibration model on avg@16.

| Model | Method | AIME | AMC |
|---|---|---|---|
| Qwen2.5-Math-1.5B | Internal | 16.7 | 46.8 |
| (Math Base Model) | External | 12.5 | 47.0 |
| Qwen2.5-7B | Internal | 23.3 | 55.4 |
| (Vanilla Base Model) | External | 23.3 | 55.4 |
| LLaMA3.1-8B-Instruct | Internal | 13.3 | 34.8 |
| (Instruct Model) | External | 10.0 | 34.2 |

datasets. Our model was trained on data consisting of only ∼7k queries with 4 responses sampled for each, whereas Qwen2.5-Math-PRM-7B was trained on ∼500k queries with 6-8 responses each. This indicates that for our task, fine-tuning the model itself on a small-scale dataset can achieve performance comparable to that of a general-purpose math reward model trained on a much larger dataset. In the future, we will further explore the use of more powerful external calibration models, such as Qwen2.5-Math-PRM-72B or Qwen2.5-Math-RM-72B.

**The effectiveness of test-time RL.** To demonstrate the efficacy of test-time RL, we compare the performance of Qwen2.5-Math-1.5B against its instruction-tuned version and similarly-sized, RL-trained models, using the pass@1 score with greedy decoding. The results in Table 5 indicate that test-time RL with the voting-based TTRL already surpasses the heavily instruction-tuned Qwen2.5-Math-1.5B-Instruct. Building on this, our proposed DREAM further improves performance by decoupling reliability estimation from reward calibration, achieving results comparable to models trained with large-scale RL. Importantly, large-scale RL methods rely on labeled datasets, whereas such labels are unavailable in our setting. Under these constraints, DREAM delivers competitive performance with only scarce labels and relatively small-scale RL during test-time training. These findings highlight both the efficiency and practicality of test-time adaptation, underscoring the broad applicability of DREAM comprehensively.

Table 5: Pass@1 comparison on AIME 2024 and AMC against similar models trained on large-scale labeled data. All results are based on greedy decoding, with baselines sourced from Dr. GRPO.

| Model | AIME 2024 | AMC | MATH-500 | Avg. |
|---|---|---|---|---|
| Qwen2.5-Math-1.5B | 20.0 | 32.5 | 36.2 | 29.6 |
| w/ TTRL | 16.7 | 48.2 | 72.4 | 45.8 |
| w/ DREAM | 20.0 | 50.6 | 73.8 | 48.1 |
| Qwen2.5-Math-1.5B-Instruct | 10.0 | 48.2 | 74.2 | 44.1 |
| DeepSeek-R1-Distill-1.5B-@3k | 2.5 | 21.7 | 52.2 | 25.5 |
| DeepSeek-R1-Distill-1.5B-@8k | 20.0 | 49.4 | 77.4 | 48.9 |
| Oat-Zero-1.5B | 20.0 | 53.0 | 74.2 | 49.1 |

## 5 CONCLUSION

In this work, we introduce DREAM, a novel reward calibration framework that decouples reward estimation from reinforcement learning to address the challenge of unreliable rewards on unlabeled data. To disentangle reward estimation from reinforcement learning, DREAM introduces an auxiliary judge model trained to assess the consistency between generated responses and ground-truth answers. By incorporating the judge model to calibrate voting-based rewards, DREAM enables more accurate policy optimization without explicit supervision. Extensive experiments on multiple mathematical reasoning benchmarks and diverse LLM backbones demonstrate the generalizability of our approach. We believe DREAM opens up promising directions for advancing test-time scaling of LLMs, particularly in scenarios where labeled data is scarce or unavailable.

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

## A  DETAILS OF JUDGE MODEL

### A.1  TRAINING DATA CONSTRUCTION

The training data for our judge model was constructed using the training split of the MATH dataset as the source of labeled data. Each sample in this dataset contains a mathematical problem (question) and its corresponding golden answer. For each question, we employed our backbone model to generate four diverse candidate responses. To encourage diversity, we used nucleus sampling with a temperature of 0.6 and a top-p value of 0.9. Each of the generated responses was then automatically evaluated for correctness by comparing its final answer to the golden answer from the MATH dataset. A binary score was assigned: a score of 1 indicates a correct response, while a score of 0 indicates an incorrect one. This procedure resulted in a dataset comprising ∼26,000 training samples and ∼3,000 test samples. Each sample is a triplet containing the original question, a model-generated response, and its assigned binary correctness score.

### A.2  MODEL ARCHITECTURE

We fine-tune our backbone model itself to act as the judge model for the subsequent test-time RL, predicting a scalar score for a given question-response pair. In this paper, we consistently use the judge model built upon Qwen2.5-Math-1.5B to reduce computational overhead. Specifically, we initialized the model from the pretrained backbone model. We then introduced a new special token, `<|judge|>`, into the tokenizer's vocabulary. This token is appended to the input sequence to signal the model to perform the evaluation task. To produce a fixed-size representation of the entire input sequence, we employed an attention pooling strategy, which uses a learnable query vector to compute an attention-weighted average of the transformer's final hidden states. This allows the model to dynamically determine which tokens are most relevant for the final score prediction. Finally, the pooled hidden representation is fed into a regression head to predict the final score. This head is a two-layer MLP with a SiLU activation function. The hidden dimension of the MLP is set to twice the model's hidden size, and a dropout rate of 0.05 is applied for regularization.

Table 6: Hyperparameters for Judge Model Training.

| Parameter Category | Value |
|---|---|
| **Model Configuration** | |
| Base Model | Qwen2.5-Math-1.5B |
| Pooling Strategy | Attention Pooling |
| Regression Head | 2-layer MLP |
| Head Hidden Multiplier | 2 |
| Head Dropout | 0.05 |
| **Data Configuration** | |
| Training Samples | ∼26,000 |
| Evaluation Samples | ∼3,000 |
| Max Sequence Length | 4096 |
| **Optimizer & Scheduler** | |
| Optimizer | AdamW |
| Learning Rate | $2 \times 10^{-5}$ |
| LR Scheduler | Cosine Annealing |
| Warmup Ratio | 0.03 |
| Weight Decay | 0.01 |
| **Training Strategy** | |
| Training Epochs | 3 |
| Per-Device Train Batch Size | 4 |
| Per-Device Eval Batch Size | 8 |
| Gradient Accumulation Steps | 1 |
| Loss Function | BCE Loss |

### A.3  TRAINING HYPERPARAMETERS

The model was trained using the hyperparameters specified in Table 6. We use AdamW for optimization with an initial learning rate of $2 \times 10^{-5}$ that followed a cosine-annealing decay schedule with a warm-up ratio of 0.03 and a weight decay coefficient of 0.01. The model was trained for 3 epochs with a per-device mini-batch of 4. Binary cross-entropy (BCE) loss was used as the training objective.

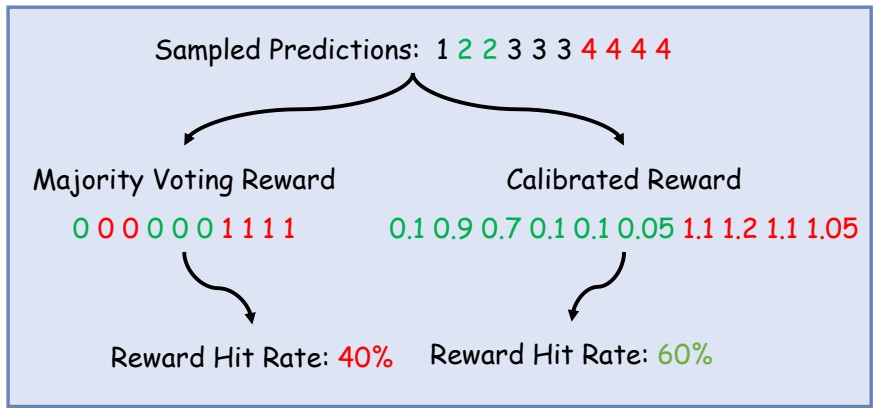

Figure 3: A simple case study of reward calibration.

## B  A SIMPLE CASE STUDY

Figure 3 provides a simple case study illustrating the critical limitation of voting-based reward mechanisms and the subsequent benefit of our DREAM's decoupled calibration. In this scenario, the policy model's sampled predictions are systematically biased, with the incorrect answer '4' appearing most frequently. A conventional majority voting approach, as employed in TTRL, would consequently assign a positive reward exclusively to the erroneous majority answer while penalizing all other candidates, including the correct answer '2'. This creates a perverse incentive structure, initiating a vicious cycle of error accumulation where the policy is guided to reinforce its own most frequent mistake. The resulting reward signal is highly misleading, as evidenced by a low "Reward Hit Rate" of 40%, which provides a poor and often detrimental gradient for policy optimization.

In contrast, DREAM effectively mitigates this failure mode by decoupling reward estimation from the policy model's consensus. By incorporating an independent reliability score from the judge model, the final calibrated reward corrects for the bias of majority voting. As shown, the correct but infrequent answers '2' is able to receive significant reward scores (0.9 and 0.7) despite their zero voting-based reward. This crucial step recovers the learning signal for the correct reasoning path, which would have otherwise been lost. Consequently, the calibrated reward structure is far more aligned with the ground truth, increasing the effective hit rate to 60% and guiding the reinforcement learning process toward a more accurate policy, rather than merely a more confident one.

Furthermore, the strength of this decoupled approach is not over-rely on a perfectly omniscient judge model. Even a moderately effective judge, one that is not flawless but is better at evaluation than the policy model is at generation, provides substantial benefits. The key advantage lies in introducing a source of evaluation whose errors are likely uncorrelated with the policy model's systemic biases. For instance, even if a moderately accurate judge assigned a slightly lower score to the correct answer '2' (e.g., 0.6, even 0.4) and a small non-zero score to an incorrect answer '3' (e.g., 0.2 or 0.1), the final calibrated reward for '2' would still be substantially higher than that for other incorrect, non-majority answers like '1' and '3'. The judge's primary role is thus to break the "tyranny of the majority" and create a more nuanced relative reward landscape. This allows group-wise optimization methods like GRPO to better distinguish between varying degrees of correctness among the candidates, preventing the policy from collapsing into a confident but fundamentally flawed state.

## C  THE USE OF LARGE LANGUAGE MODELS

In this manuscript, we use LLMs to help polish writing at the sentence level and check grammar.

