# OpenReview forum: "DREAM: Decoupled Reinforcement Learning with Reward Measurement for Large Language Model Test-time Training"
_ICLR.cc/2026/Conference — ICLR 2026 Conference Withdrawn Submission_

### Official Review · Reviewer_3iZk · 2025-10-26

**Soundness:** 1
**Presentation:** 2
**Contribution:** 2
**Rating:** 2
**Confidence:** 3

**Summary:**

This paper addresses the problem of enhancing LLMs through test-time training using unlabeled data. The authors propose DREAM, which decouples reward estimation from reinforcement learning via an LLM-based calibration model trained on a reference dataset. The method integrates calibrated scores into voting-based reward estimation to reduce bias and improve reliability. The authors conduct numerical experiments on benchmark datasets to evaluate DREAM.

**Strengths:**

This paper attempts to enhance reinforcement learning in large language models (LLMs) without relying on labeled data.

**Weaknesses:**

The proposed method appears to lack sufficient novelty. The main technique is the use of reference data to learn a calibration score. However, designing a canonical answer extraction function to compute this calibration score seems non-trivial—especially for complex tasks—and the function may not always be correct. Moreover, such a function would likely need to be redefined for each new task.

The method shares the same name as another work, “DREAM: Decoupled Discriminative Learning with Bigraph-aware Alignment for Semi-supervised 2D-3D Cross-modal Retrieval,” which may cause confusion.

The empirical improvements also appear marginal. For instance, in Figure 2(a), the improvement is less than 2 in three out of four cases, and in Figure 2(b), the performance even drops.

**Questions:**

In Equation (5), why are the two terms simply added? If one term is more accurate, wouldn’t it make sense to rely primarily on that term? Additionally, what happens if the two terms are highly correlated or even identical? Are there any theoretical guarantees in such cases?

What is the relationship between this method and the semi-supervised RL approach proposed in https://arxiv.org/abs/1612.00429? Why was this comparison not included?

---

### Official Review · Reviewer_ZULn · 2025-11-01

**Soundness:** 3
**Presentation:** 3
**Contribution:** 3
**Rating:** 6
**Confidence:** 4

**Summary:**

In this paper, the authors focus on LLMs' TTRL in unlabeled data. Specifically, they focus on the error accumulation based on the unreliable reward signals generated by the majority voting mechanism. To solve this problem, the authors proposed an innovative framework called DREAM. The core innovation of this model is that it decouples the reward evaluation from the policy optimization process. The experimental results show that this method outperforms various baselines.

**Strengths:**

1. The experiments in this paper are convincing. The authors validate the effectiveness of the method on various LLMs and three different mathematical reasoning tasks. They also provide an ablation study and cross-dataset validation. In general, I believe the experiments fully support the paper.

2. The structure of this paper is clear, and the writing is well-formatted. The authors provide sufficient details and figures/tables to better clarify their idea.

**Weaknesses:**

1. The motivation of this paper is not clear enough in the introduction section. The author may consider adding an example or case study to enhance the readability.

2. The judge model doesn't provide sufficient details, including its architecture, training methods, and other key information. Although the authors further clarify this information in the appendix, the lack of this information weakens the completeness of the paper. The paper should cover all necessary details without the appendix.

**Questions:**

1. The DREAM framework relies on a labeled reference dataset to train the judge model. An open question is how sensitive the framework is to the dataset quality. In some cases, the available reference dataset may be small and noisy. Can DREAM be applied in these cases and maintain its effectiveness?

2. The method requires multiple samples at test time. This strategy may increase the computational cost. An open question is how the author balances the trade-off between performance gain and computational cost. Are there systematic methods to reduce the number of samples without sacrificing performance?

---

### Official Review · Reviewer_xzVK · 2025-11-01

**Soundness:** 2
**Presentation:** 3
**Contribution:** 2
**Rating:** 4
**Confidence:** 4

**Summary:**

This paper presents DREAM (Decoupled Reinforcement Learning with Reward Measurement), a framework designed to improve large language model test-time training on unlabeled data. The authors address a key limitation in existing test-time reinforcement learning approaches, where majority voting-based reward estimation can be inaccurate and lead to biased policy optimization. DREAM introduces a two-stage approach: first training an auxiliary judge model on reference data to assess response reliability, then using these calibration scores to refine majority voting rewards during policy optimization. The framework is evaluated on mathematical reasoning benchmarks (AIME 2024, AMC, MATH-500) across multiple backbone models, showing improvements over the baseline TTRL method. The core contribution lies in decoupling reward estimation from policy learning to mitigate error accumulation, though the improvements appear modest and the approach requires additional computational overhead for training the judge model.

**Strengths:**

1. The authors' approach of employing an additional independent judge model to calibrate the reliability of voting-based results is conceptually interesting and demonstrates notable innovation in its motivation.
2. The authors demonstrate strong experimental performance, achieving significant improvements across multiple models and datasets.

**Weaknesses:**

1. The proposed method appears limited to tasks with clear, verifiable answers, making it primarily applicable to reasoning domains like mathematics and coding rather than serving as a general post-training approach for LLMs. Many real-world applications involve subjective or open-ended tasks without definitive correct answers, where the binary correctness assessment underlying the judge model becomes impractical or impossible to implement.
2. Regarding Equation 3, it's questionable why the authors employ mean squared error (MSE) loss when the labels are binary. Since the judge model is essentially performing binary classification (correct vs. incorrect), a classification loss such as cross-entropy loss with temperature scaling would be more theoretically appropriate and potentially more effective than treating this as a regression problem with MSE loss.
3. A critical concern is that the judge model's performance should significantly impact the final LLM performance, yet the experiments fail to demonstrate how variations in judge model accuracy affect the overall results.
4. The judge model itself may suffer from bias or overconfidence issues, as it is trained on existing datasets with fixed parameters. When encountering new samples during the training process, these inherent biases could potentially propagate and negatively impact the LLM's performance.

**Questions:**

see weaknesses.

---

### Official Review · Reviewer_eSoi · 2025-11-02

**Soundness:** 3
**Presentation:** 3
**Contribution:** 3
**Rating:** 4
**Confidence:** 4

**Summary:**

The authors propose DREAM, which decouples reward estimation from reinforcement learning to generate more reliable rewards. Unlike previous approaches that rely on majority voting from model outputs, DREAM trains a separate judge model to evaluate the reliability of generated answers. These calibrated rewards are then used to guide policy optimization. The authors demonstrate DREAM's effectiveness across multiple backbone models and mathematical reasoning benchmarks, showing consistent performance improvements over existing methods.

**Strengths:**

- The paper is well-organized and easy to follow.
- The motivation for the research is sound.
- The experimental setup and ablation studies are comprehensive.

**Weaknesses:**

- According to Table 1 and Figure 2, performance gains introduced by DREAM are limited compared to TTRL.
- It remains unclear how much GPU runtime was required for experiments.

**Questions:**

- In terms of training efficiency, it's unclear whether DREAM is more efficient than TTRL, as the paper lacks a detailed comparison of training time requirements between these two approaches.
- When using Qwen-2.5-PRM as an external reward model for training LLaMA3.1-8B-Instruct in Table 4, could there be a potential distribution shift between these different models that impacts the reliability of calibration scores? Is this potential distribution shift responsible for the underperformance of the external approach observed in the results?

---

### Note · Authors · 2026-01-06

I have read and agree with the venue's withdrawal policy on behalf of myself and my co-authors.